# Quantifying antibody kinetics and RNA detection during early-phase SARS-CoV-2 infection by time since symptom onset

**Benny Borremans[1,2,3]\*, Amandine Gamble[1], KC Prager[1], Sarah K Helman[1], Abby M McClain[4], Caitlin Cox[1], Van Savage[1,5], James O Lloyd-Smith[1]**

[1]Ecology and Evolutionary Biology Department, University of California, Los Angeles, Los Angeles, United States; [2]I-BioStat, Data Science Institute, Hasselt University, Hasselt, Belgium; [3]Evolutionary Ecology Group, University of Antwerp, Antwerp, Belgium; [4]National Marine Mammal Foundation, San Diego, United States; [5]Biomathematics Department, University of California, Los Angeles, Los Angeles, United States

**Abstract** Understanding and mitigating SARS-CoV-2 transmission hinges on antibody and viral RNA data that inform exposure and shedding, but extensive variation in assays, study group demographics and laboratory protocols across published studies confounds inference of true biological patterns. Our meta-analysis leverages 3214 datapoints from 516 individuals in 21 studies to reveal that seroconversion of both IgG and IgM occurs around 12 days post-symptom onset (range 1–40), with extensive individual variation that is not significantly associated with disease severity. IgG and IgM detection probabilities increase from roughly 10% at symptom onset to 98–100% by day 22, after which IgM wanes while IgG remains reliably detectable. RNA detection probability decreases from roughly 90% to zero by day 30, and is highest in feces and lower respiratory tract samples. Our findings provide a coherent evidence base for interpreting clinical diagnostics, and for the mathematical models and serological surveys that underpin public health policies.

**\*For correspondence:**
bennyborremans@gmail.com

**Competing interests:** The authors declare that no competing interests exist.

## Introduction

Since its emergence in December 2019, the SARS-CoV-2 pandemic has been the subject of intense research assessing all facets of the pathogen and its rapid global spread. Serology – the measurement of serum antibodies – provides crucial data for understanding key aspects of infection and epidemiology (*Weitz et al., 2020*). At the level of populations, serologic data can provide insights into virus spread by enabling estimation of the overall attack rate, and seroprevalence estimates can elucidate the potential for herd immunity (*Stringhini et al., 2020*; *Bryant et al., 2020*). In addition, these estimates are essential for developing accurate mathematical models of virus transmission dynamics, which provide the foundation for policies to reopen societies (*Krsak et al., 2020*; *Angulo et al., 2020*; *Kissler et al., 2020*). At the level of individuals, the presence and concentration of antibodies against SARS-CoV-2 are indicators of past exposure, providing insights over a much wider temporal window than other metrics. When considered jointly with PCR testing to detect viral RNA, antibodies substantially improve the probability of detecting present and past infections (*Prager et al., 2019*). This improvement is highly valuable because RNA detection is typically limited to a relatively brief period of infection, and because PCR sensitivity varies considerably with infection severity and biological sample type (*Azkur et al., 2020*; *Yongchen et al., 2020*). Assessment of the levels of different antibody types (e.g. IgG, IgM) may even be used to infer approximately when individuals became infected (*Azkur et al., 2020*; *Chang et al., 2005*; *Du et al., 2020*; *Borremans et al.,*

*2016*), while detection of neutralizing antibodies may indicate protection from reinfection (*Ni et al., 2020*).

These applications of serologic data depend critically on knowing when different antibodies against the pathogen become detectable (seroconversion time), how their concentrations change over time (antibody level kinetics) and how long they last (antibody decay) (*Lipsitch et al., 2020*). When these key factors are known, serologic data become a powerful tool for inferring infection attack rate and transmission dynamics in the population (*Bryant et al., 2020*; *Winter and Hegde, 2020*). Five months into the pandemic, a remarkable number of serologic studies on the initial immune response against SARS-CoV-2 had been published. These studies were conducted in different laboratories, used different assays and sampling methods, and were performed on different patient groups that showed different clinical manifestations of SARS-CoV-2 infection (*Whitman et al., 2020*; *Lassaunière et al., 2020*; *Kontou et al., 2020*).

This extensive variation arising from many sources creates substantial challenges for integrating existing data into one coherent picture of antibody kinetics and viral RNA detection following SARS-CoV-2 infection. In 21 studies reporting the kinetics of anti-SARS-CoV-2 antibodies, we found the use of 8 different antibody assays, 10 different target antigens, and 9 different reported antibody level units (studies are listed in the Materials and methods section). Additionally, the temporal resolution at which studies collect data is highly variable: while some studies report antibody measurements for specific days, many bin results into periods of multiple days or even weeks. Integrated analysis of such diverse data is challenging, and requires statistical methods specifically developed for this purpose. Yet this type of integration is essential to capitalize on the limited and precious data available, to assess to what degree antibody and RNA detection patterns are affected by assay type and target antigen choice, and to establish consensus patterns. For example, a properly integrated analysis would better enable us to test whether antibody patterns depend on disease severity (*Huang et al., 2020*; *Tan et al., 2020*).

In this study, we quantified IgG and IgM antibody kinetics and RNA detection probability during SARS-CoV-2 infection (up to 60 days post-symptom onset) by aggregating data from published sources. We formally characterized IgG and IgM seroconversion times, detection probabilities over time and antibody level kinetics using methods tailored to accommodate the diverse ways in which data have been collected and reported. We investigated how these variables are affected by disease severity, assay type and targeted antigen, and how patterns differ between IgG and IgM. We also assessed how antibody level kinetics relate to the probability of detecting viral RNA in various biological samples. We estimated mean values as well as observed variation of all variables in order to provide the complete picture required to interpret serological and RNA testing data, inform mitigation strategies and parameterize mathematical models of pathogen transmission while accounting for variability. This formal integration approach enabled us to leverage 3214 data points from 516 individuals with symptoms ranging from asymptomatic to critical, published in 21 studies, resulting in a quantitative synthesis of diverse data on anti-SARS-CoV-2 antibody patterns and RNA detection during the early phase of infection.

## Results

### Data extraction

We extracted data from 21 preprints and peer-reviewed articles reporting data on SARS-CoV-2 RNA or IgG, IgM or neutralizing antibodies against the virus in humans (see Materials and methods). When available, disease severity information was classified into three groups: asymptomatic/subclinical (n = 11 individuals), mild/moderate (n = 166), and severe/critical (n = 58). Unfortunately, the sample size for the asymptomatic group was too low for quantitative analyses. For 359 individuals, insufficient data were available for disease severity categorization, and these individuals were therefore excluded from analyses of the impact of disease severity. Published results were variously reported as exact days, intervals up to 22 days, or mean times for multiple individuals, while test results were reported as values for one individual or mean values for multiple individuals. Data after 30 days post-symptom onset were particularly underrepresented, but included because in aggregate they provide key insights. When reporting enzyme-linked immunosorbent assay (ELISA) results in the main text, IgG results are shown for assays targeting the nucleoprotein (NP) antigen (ELISA-NP), and

IgM results are shown for assays targeting the Spike antigen (ELISA-Spike; whole or subunit), as these assays are most often used for the two antibody types (*To et al., 2020*; *Sethuraman et al., 2020*). Results for other assays and antigens are shown in *Figure 1—figure supplements 1* and *2*.

## The distribution of seroconversion times

Stepwise bootstrapping was used to estimate seroconversion times, using 270 data points from 99 individuals for IgG and 240 data points from 71 individuals for IgM. Mean IgG seroconversion time is 13.3 days post-symptom onset when using ELISA-NP and 12.6 for IgM using ELISA-Spike (*Figure 1a*). These results do not differ significantly (t = 0.22, df = 7.7, p=0.84) and are similar for magnetic chemiluminescence enzyme immunoassay (MCLIA; *Figure 1b*). Variation in seroconversion times is substantial regardless of assay, for both IgG (sd = 5.7) and IgM (sd = 5.8).

Disease severity does not significantly affect seroconversion time, for IgM or for IgG (*Figure 1c–d*). Mean IgM seroconversion time for mild/moderate cases is 12.3 days post-symptom onset vs 13.2 for severe/critical cases (t = −0.2, df = 23.5, p=0.83). Mean IgG seroconversion time for mild/moderate cases is 12.9 days post-symptom onset, vs 15.5 for severe/clinical cases (t = −0.96, df = 14.8, p=0.35). A detailed overview of seroconversion time results including means and standard deviations is provided in *Figure 1—figure supplements 3–5* and *Figure 1—source data 1*.

## Antibody detection probability

While estimates of seroconversion time provide information about the first moment at which antibodies can be detected, changes in detection probability over time provide useful information about the proportion of individuals that has detectable antibodies, and hence the expected test sensitivity at the population scale. Sample sizes for these analyses (see Materials and methods) are 8053 data points for IgG and 7935 for IgM, with daily mean sample sizes of 224 and 220, respectively. The probability of detecting IgG (ELISA-NP) increases over time, reaching a maximum around 25–27 days post-symptom onset, at which point between 98% and 100% of individuals test positive (*Figure 2a*). Detection probability remains at this maximum level for the remainder of the days available in the studies existing at the time of writing (up to 60 days for ELISA-Spike, *Figure 2—figure supplement 1*). This pattern is consistent across assays (*Figure 2—figure supplement 1*). IgM (ELISA-Spike) detection probability is similar to that of IgG until its peak near 90% around 23–25 days post-symptom onset, after which it starts to decrease, reaching roughly 65% detection probability around 60 days post-symptom onset (*Figure 2*, *Figure 2—figure supplement 2*). Although data on neutralizing antibody presence were sparse, we observe that detection probability rapidly rises to near 100%, where it remains up to the last time available in the dataset (*Figure 2—figure supplement 3*; 29 days post-symptom onset). Patterns in detection probability do not differ significantly between mild/moderate and severe/clinical cases, aside from a slightly steeper rise for severe/critical cases (*Figure 2—figure supplement 4*). *Figure 2—source datas 1–2* provide the estimated detection probabilities over time for IgG and IgM.

## RNA detection probability

Samples sizes for observed and interpolated data are 7443 and 1793 for upper and lower respiratory samples and 1179 for fecal samples, with mean daily sample sizes of 226, 72 and 39, respectively. The probability of detecting viral RNA in respiratory and fecal samples is high (80–100%) at symptom onset and is consistently highest for lower respiratory tract samples (*Figure 2b*). Detection probability decreases rapidly at rates dependent on sample type, and most rapidly for upper respiratory tract samples, but the proportion of positive samples approaches zero around 30 days post-symptom onset for each sample type. Raw RNA detection probability data are provided in *Figure 2—source datas 3–5*.

## Antibody level kinetics

Antibody kinetics were analyzed by fitting a Gompertz growth rate function using Bayesian MCMC. While all subsets of the data were fit well by this model, we found some differences in antibody level kinetics depending on antibody, assay and antigen (*Figure 3A*). Full model fitting results for each assay can be found in Appendix 1.

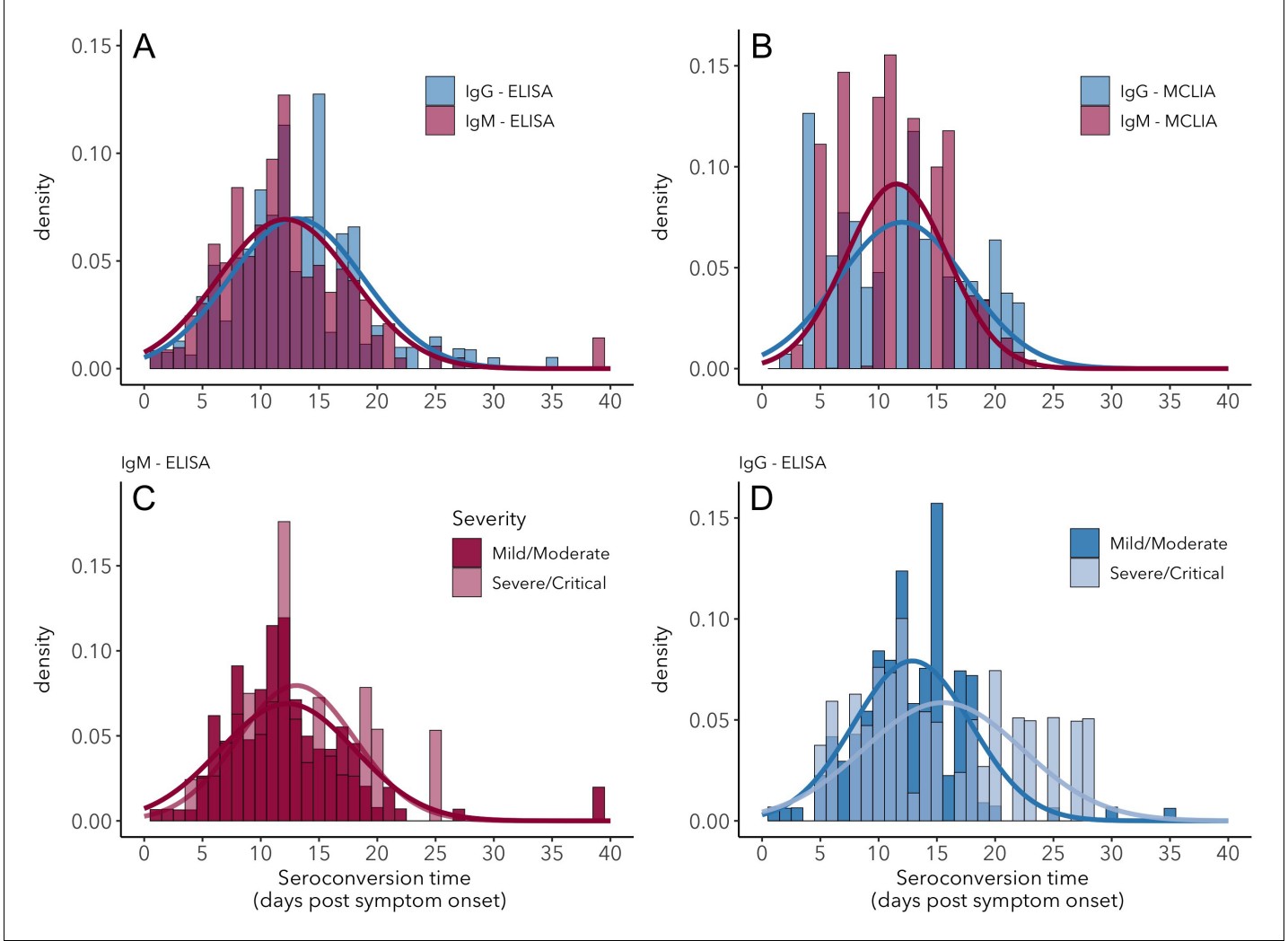

**Figure 1.** Seroconversion time distributions for IgG and IgM. (**A**) IgG and IgM detected using ELISA. (**B**) IgG and IgM detected using MCLIA. (**C**) IgM and (**D**) IgG seroconversion related to disease severity. IgG and IgM ELISA results are shown for the NP and Spike antigens, respectively, because these had the largest sample sizes. Lines indicate fitted normal distributions.

The online version of this article includes the following source data and figure supplement(s) for figure 1:

**Source data 1.** Fitted normal distribution parameters for seroconversion time using different assays.

**Figure supplement 1.** Distributions of IgG and IgM seroconversion times (including all assays) for increasing levels of data inclusion, from exact time data only (time period = 0) to the inclusion of the longest reported time periods (all).

**Figure supplement 2.** IgG seroconversion time distributions for different assays.

**Figure supplement 3.** IgG seroconversion time distributions for different target antigens.

**Figure supplement 4.** IgM seroconversion time distributions for different assays.

**Figure supplement 5.** IgM seroconversion time distributions for different target antigens.

Peak antibody level is reached around days 14–20 post-symptom onset, and the timing depends on antigen: both IgG and IgM peak levels are reached earlier when measured using ELISA NP than when using ELISA Spike (ELISA NP mean = 14.3 days, 95% CrI 12.0–16.1; ELISA Spike mean = 20.0 days, 95% CrI 17.6–22.4; 95% CrI for the difference = 2.7 to 9.2). The peak timing does not differ significantly between IgG and IgM when both are measured using ELISA Spike (IgG mean = 20.4 days, 95% CrI 16.8–24.1; IgM mean = 19.1 days, 95% CrI 15.6–22.4; 95% CrI for the difference = −6.4 to 3.5), nor when using ELISA NP (IgG mean = 15.2 days, 95% CrI 12.8–17.2, IgM mean = 12.2, 95% CrI 7.8–16.2; 95% CrI for the difference = −1.8 to 7.8). All estimates and pairwise

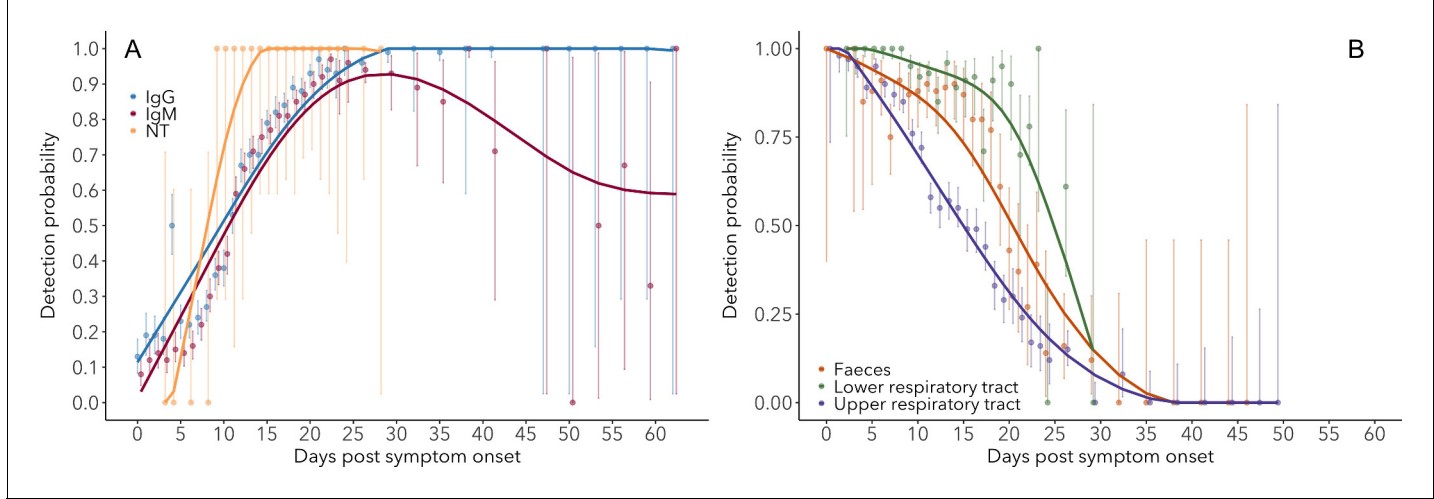

**Figure 2.** Detection probability of IgG, IgM and NT (neutralizing) antibody (**A**) and RNA in different sample types (**B**) over time since symptom onset. Points are mean values for each day. Bold lines are flexible smoothed splines fit to the data. Error bars indicate binomial exact 95% confidence intervals of the mean, based on daily sample size. Note that error bars after day 30 tend to be large, due to the limited available data. IgG and IgM values are those detected using any assay/antigen. After day 25, results are pooled into 3-day periods in order to improve estimates.

The online version of this article includes the following source data and figure supplement(s) for figure 2:

**Source data 1.** IgG (ELISA-NP) detection probability.
**Source data 2.** IgM (ELISA-Spike) detection probability.
**Source data 3.** RNA – upper respiratory tract detection probability.
**Source data 4.** RNA – lower respiratory tract detection probability.
**Source data 5.** RNA – feces detection probability.
**Figure supplement 1.** IgG detection probability for different assays/antigens.
**Figure supplement 2.** IgM detection probability for different assays/antigens.
**Figure supplement 3.** Detection probability for serum IgG (ELISA-NP) and IgM (ELISA-Spike), and for viral RNA in upper respiratory tract samples.
**Figure supplement 4.** Detection probability for serum IgG and IgM in mild/moderate and severe/critical cases.

statistics, including those for antibody levels measured using MCLIA, are shown in *Figure 3—source datas 1–2*.

Antibody growth rates measured using ELISA NP tend to be higher than those measured using ELISA Spike (ELISA NP mean = 0.55/day, 95% CrI 0.48–0.64; ELISA Spike mean = 0.39/day, 95% CrI 0.34–0.44; 95% CrI for the difference = 0.07 to 0.26). The rate of increase for IgM does not differ significantly from that of IgG when both are measured using ELISA Spike (IgG mean = 0.39/day, 95% CrI 0.32–0.46; IgM mean = 0.41/day, 95% CrI 0.34–0.49; 95% CrI for the difference = −0.08 to 0.13), nor when measured ELISA NP (IgG mean = 0.53/day, 95%CrI 0.45–0.61; IgM mean = 0.68/day, 95% CrI 0.42–1.03; 95% CrI for the difference = −0.50 to 0.14). All estimates and pairwise statistics, including those for antibody levels measured using MCLIA, are shown in *Figure 3—source datas 3–4*.

Disease severity does not significantly affect the time at which peak levels are reached for IgG (*Figure 3B*; mild mean = 14.0 days, 95% CrI 10.8–17.1; severe mean = 15.9 days, 95% CrI 10.7 to 20.6; 95% CrI for the difference = −8.0 to 4.2). However for IgM, peak antibody levels are reached approximately 7.0 days earlier for mild cases than severe cases (*Figure 3C*; mild mean = 15.6 days, 95% CrI 12.8–19.0; severe mean = 22.7 days, 95% CrI 18.5–26.6; 95% CrI for the difference = −12.2 to −1.8). Corresponding patterns are observed for antibody growth rate, which does not differ between mild and severe cases for IgG (mild mean = 0.58/day, 95% CrI 0.45–0.72; severe mean = 0.51/day, 95% CrI 0.36–0.69; 95% CrI for the difference = −0.16 to 0.28), but does for IgM, with levels increasing more rapidly for mild cases (mild mean = 0.51/day, 95% CrI 0.42–0.60; severe mean = 0.34/day, 95% CrI 0.28–0.42; 95% CrI for the difference = 0.05 to 0.28).

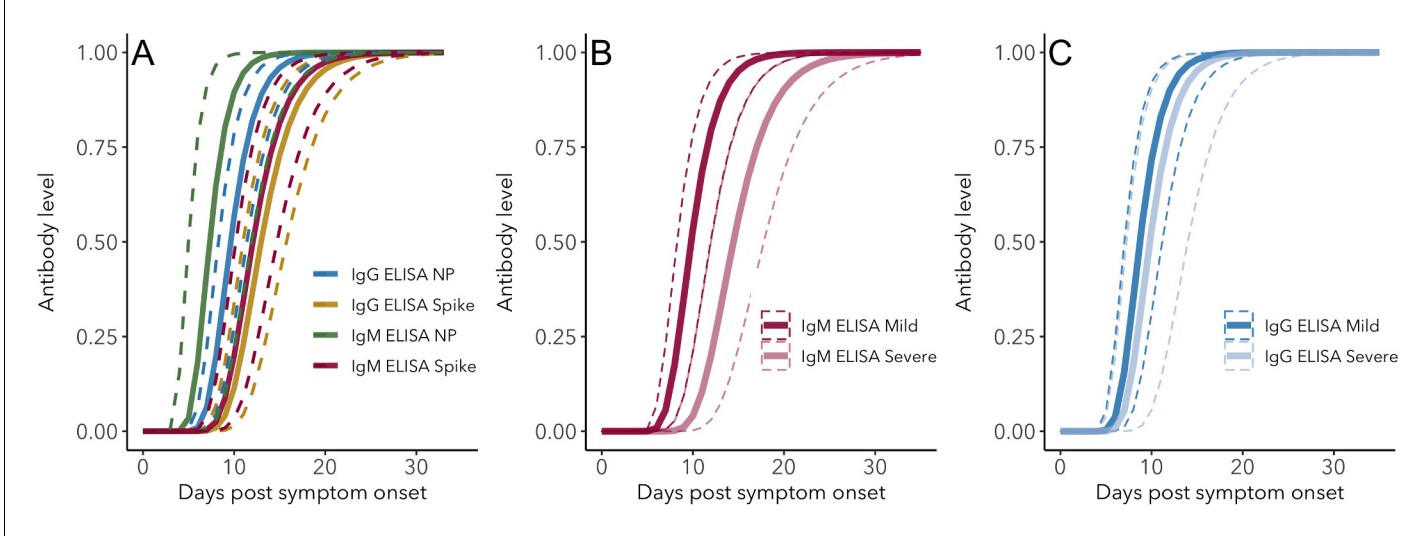

**Figure 3.** IgG and IgM antibody level kinetics for ELISA NP and Spike assays (**A**) and disease severity for IgM (**B**) and IgG (**C**). Measured using ELISA Spike and ELISA NP, respectively. Fitted functions use the posterior mean values for increase rate and start of the increase phase (displacement). Dotted lines show upper and lower 95% credible intervals. Note that the upper CI of IgM ELISA severe overlaps with the lower CI of mild cases, as do the upper CIs of IgG ELISA mild and severe. In order to allow the comparison of increase rate patterns, normalized peak antibody levels were set to one for all functions.

The online version of this article includes the following source data for figure 3:

**Source data 1.** Peak antibody level time posterior means and 95% credible intervals (CrI).
**Source data 2.** Peak antibody level time pairwise posterior differences.
**Source data 3.** Growth rate posterior means and 95% credible intervals (CrI).
**Source data 4.** Growth rate pairwise posterior differences.

## Discussion

By leveraging and integrating multiple data sources on key aspects of the antibody response against SARS-CoV-2, we were able to produce quantitative estimates of the mean and variation of seroconversion timing, antibody level kinetics, and the changes in antibody and RNA detection probabilities. These results provide critical reference information for serological surveys, assay sensitivity and risk of false-negative results, transmission models and herd immunity assessments. By combining data from 21 different studies using different assays, antigens, protocols and patient groups, we were able to quantify the means and, crucially, the extent of variation of important serologic and RNA detection parameters. Together, these antibody and RNA detection probability patterns provide an essential evidence base for informing sampling designs (*Table 1*). *Figure 4* provides an overview of the key patterns.

Seroconversion time is highly variable between individuals, with a mean around 12–13 days post-symptom onset. We find that IgG and IgM can be detected as early as 0 days post-symptom onset in 10–20% of patients, which indicates that seroconversion can happen at, and likely before, the onset of detectable symptoms. To our knowledge, seroconversion prior to symptom onset has not been reported, which is likely due to the fact that such cases are typically not under investigation using serologic assays. By integrating a wide range of data sources, we detect greater variation in seroconversion timing than previously observed, and importantly, it was possible to quantify the distributions around the mean seroconversion times (*Huang et al., 2020*; *Zhao et al., 2020*; *Haveri et al., 2020*).

Patterns of IgM and IgG detection align with immunological expectations, as IgM antibodies are typically present during the early phase of the immune response, while IgG antibodies remain detectable for much longer periods (*Xiao et al., 2020*). We detected IgG and IgM antibodies in nearly all (98–100%) individuals by days 22–23 post-symptom onset, consistent with recent findings (*Kraay et al., 2020*). While IgG detection remains at this level for at least the range of times in the

**Table 1.** Examples of how improved knowledge of antibody and RNA detection patterns can inform sampling designs.

| Question | What to test for | Optimal timing to test | Comments | Importance |
|---|---|---|---|---|
| Has an individual been exposed in the past? | IgG | 25-60(+) days post symptom onset | IgG persistence: possibly 1–2 years based on other human coronaviruses (*Chang et al., 2005*). | Transmission models (*Weitz et al., 2020*; *Kucharski et al., 2020*) Herd immunity (*Lassaunière et al., 2020*; *Theel et al., 2020*). |
| Is an individual currently infected? | Viral RNA | <30 days post-symptom onset | Preferable: sequential tests because of extensive variation in detection (*Wölfel et al., 2020*). Detection probability highest for lower respiratory tract or fecal samples, but upper respiratory tract samples are necessary to assess transmission potential. | Assess transmission risk to others; contact tracing *Giordano et al., 2020*; Parameterization of transmission models (*Weitz et al., 2020*; *Kucharski et al., 2020*). |
| How recently was an individual exposed? | IgM, IgG | >25 days post-symptom onset | IgG indicates exposure, which is more likely to be recent if IgM is also present, and longer ago if IgM is absent. | Recent exposure is more likely correlated with transmission risk, and is a useful measure for prioritizing contact tracing, notably for asymptomatic cases (*Okba et al., 2020*). |

dataset (60 days for ELISA-Spike), the proportion of IgM-positive samples decreases after roughly 28 days post symptom onset, reaching around 65% by day 60. In other words, a growing proportion of individuals loses detectable IgM from day 30 onwards. We also detect a robust effect of viral antigen, where responses against NP rise faster than those against Spike, for both IgM and IgG. The quantification of changes in detection probability over time is relevant for clinical testing and assay choice and will determine test sensitivity (*Sethuraman et al., 2020*).

It has been postulated that disease severity and humoral immunity against SARS-CoV-2 are correlated, but results so far have been inconclusive (*Okba et al., 2020*). Here, we did not detect any significant effects of disease severity on antibody patterns, with the single exception that we estimated a lower rate of IgM increase in severe/critical cases relative to mild/moderate cases. Regarding seroconversion times, an earlier study analyzed 28 cases to find that IgM seroconversion times appeared to be the same for severe and non-severe cases, but their analysis of 45 cases showed that IgG seroconversion was earlier for severe cases (*Tan et al., 2020*). Similarly, earlier seroconversion in severe cases has been observed for SARS-CoV-1 (*Lee et al., 2006*), but this result was not consistent across studies (*Chan et al., 2005*). Our findings do not support the idea that severe cases seroconvert faster. Indeed, the only significant effect of severity in our analyses is that the inferred growth rate of IgM levels is slower for severe/critical cases. It is not clear whether this reflects a relevant biological difference, considering that all other parameters do not differ among disease severity categories. The consensus patterns from our meta-analysis suggest that any interaction between disease severity and antibody response must be subtle and sensitive to other sources of variation, explaining the inconsistencies seen across studies. Note that the IgG seroconversion histogram for severe/critical cases (*Figure 1d*) appears bimodal, with fewer datapoints between 13 and 18 days post-symptom onset. This could either be a consequence of an underrepresentation of these times in the different studies or a signal of a true underlying pattern, but unfortunately the data to distinguish between these two hypotheses are not currently available.

Given the finding that disease severity does not have major effects on early-phase antibody patterns, and assuming no cryptic relationship between severity and the factors governing protective immunity, then mild cases could be substantial contributors to the development of herd immunity development. This finding may also be important for vaccine efficacy; however, it is not yet known whether the presence of IgG or IgM correlates with protective immunity (*Altmann et al., 2020*), although we do observe a similar pattern for neutralizing antibody detection (*Figure 2a*).

The extensive individual variation in antibody patterns, which is a common phenomenon across many viral infections (*Pacis et al., 2014*), may affect the accuracy of transmission models (*Weitz et al., 2020*). For example, if seroconversion times reflect the actual end of infectiousness and onset of immunity (i.e. the transition from Infectious to Removed in SEIR-type models *Li et al., 2020*), the observed range of 0 to 40 days post-symptom onset may need to be represented in the infectious period distribution. It is important to carefully consider how this variation may affect

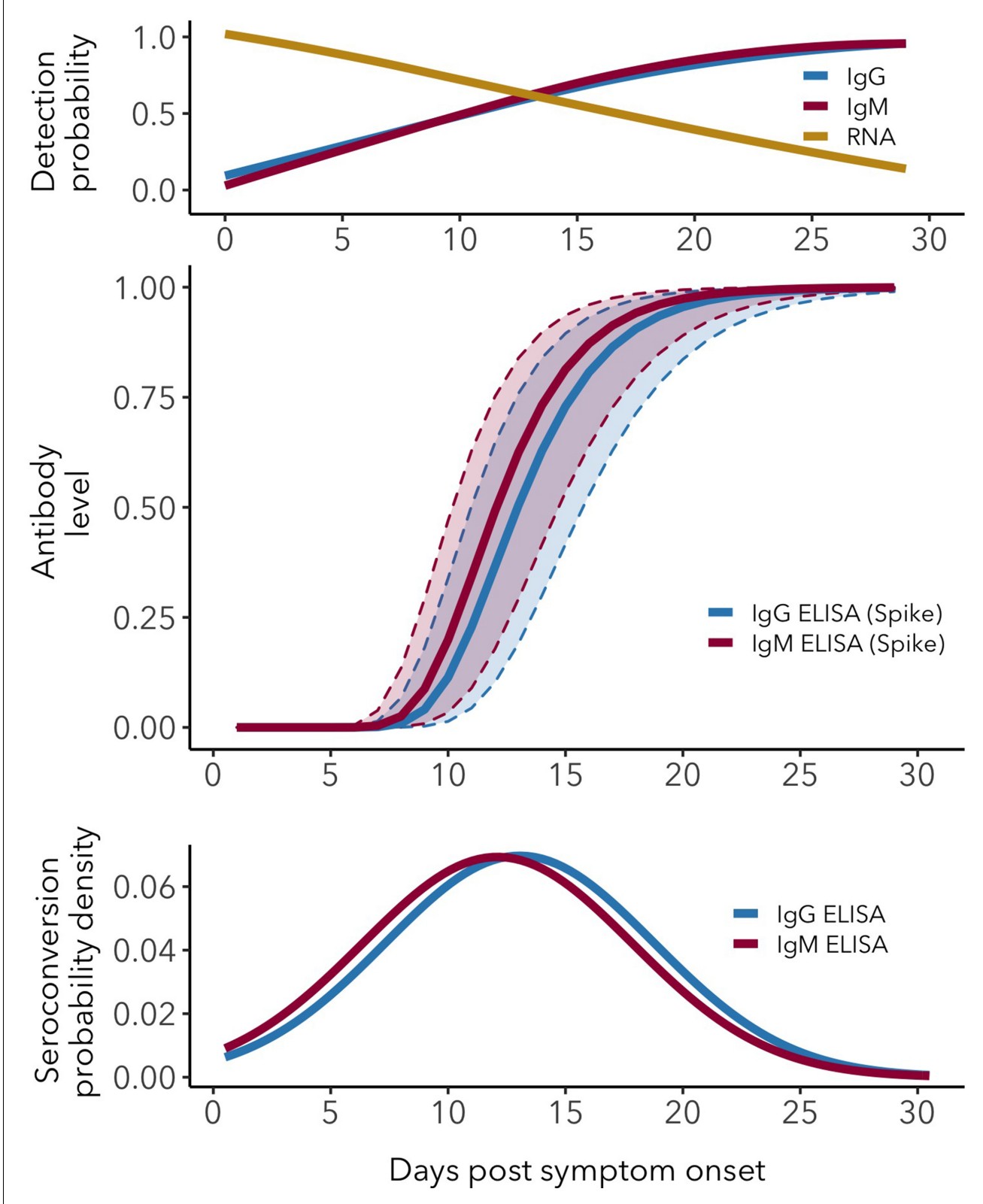

**Figure 4.** Antibody and RNA detection patterns during the early phase of SARS-CoV-2 infection. (Top) Fitted splines for the detection probabilities of serum IgG and IgM (measured using any assay/antigen), and of RNA in upper respiratory tract samples. (Middle) Modeled IgG and IgM level kinetics with 95% credible intervals, with normalized peak antibody levels set at one to allow direct comparison of growth rates. (Bottom) Estimated distribution of observed IgG and IgM seroconversion times.

model conclusions, and whether it should be taken into account explicitly (*Wearing et al., 2005*), especially given the heavy reliance of policy-makers on COVID-19 transmission models (*Kissler et al., 2020*).

We observed clear patterns of RNA detection that have several important implications, particularly for sampling designs. First, it is clear that the probability of detecting RNA is highly dependent on sample type, consistent with previous observations (*Tan et al., 2020*; *Memish et al., 2014*). Lower respiratory tract samples have the highest probability of testing positive for SARS-CoV-2 RNA, particularly after about 15 days post-symptom onset. During the first 8 days, 100% of lower respiratory tract samples tested positive for RNA. While detection probabilities for fecal and upper respiratory tract samples are nearly this high at symptom onset, they decrease much more rapidly, with the lowest average detection probabilities for upper respiratory samples. Nevertheless, it appears that by 30 days post-symptom onset detection probability approaches zero for all sample types, although it is important to note that the dataset did not include lower respiratory samples beyond day 29, which means that the true detection endpoint in lower respiratory samples could not be determined. These results match those from multiple studies (*Tan et al., 2020*; *To et al., 2020*; *Sethuraman et al., 2020*; *Guo et al., 2020*). When interpreting results on RNA detection, it is important to note that the presence of RNA does not necessarily imply the presence of live virus (*Theel et al., 2020*; *Wölfel et al., 2020*).

One potential caveat for any analysis of data reported as time since symptom onset is that variation in the incubation period (time between infection and symptom onset) can affect the estimated timing of antibody kinetics and RNA detection. The mean incubation period is estimated to be around 7–8 days, with a standard deviation of 4.4 (*Ma et al., 2020*). The clear antibody and RNA detection patterns we observe here suggest that the effect of this variation does not obscure broad patterns, but relative results may be affected if the incubation period differs between certain groups of individuals. This could indeed be the case for disease severity, as mild cases are estimated to have a longer incubation period (8.3 days) than severe cases (6.5 days) (*Ma et al., 2020*).

In summary, this study provides an up-to-date, comprehensive reference of key antibody and RNA detection parameters, including estimates of variation that can be used to inform serological surveys and transmission models (*Table 1*). As more data on SARS-CoV-2 become available, parameters can be updated through the use of the algorithms made available in the accompanying R code.

## Materials and methods

### Article selection

We considered preprints and peer-reviewed articles reporting the presence (positive or negative) or levels for IgG, IgM or neutralizing antibodies against SARS-CoV-2 or SARS-related CoV RaTG13 measured by enzyme-linked immunosorbent assay (ELISA), magnetic chemiluminescence enzyme immunoassay (MCLIA), lateral flow immunoassay (LFIA) or plaque reduction neutralization test (PRNT). In addition, we considered studies reporting PCR data from various biological samples, based on various PCR protocols. To be included in the study, we required that data were associated with information about time since symptom onset at the moment of sample collection. The search terms 'SARS-CoV-2' and 'COVID-19' were used in combination with the following search terms: serolog*, antibod*, IgG, IgM, RNA, shedding. This strategy was used in the databases Google Scholar, Pubmed and medRxiv. This resulted in about 850 candidate articles and preprints. Within these results, a first selection of candidate articles was performed by assessing the titles, in order to filter articles containing new data (i.e. excluding reviews, opinion articles, modeling studies, etc.). This narrowed down the list of candidate articles to 37, which were screened in detail. The final selection step required articles and preprints to show raw data in tables or figures and include data on time post-symptom onset. A selection process flowchart is shown in *Figure 5*. We included articles available up to May 1 2020 that contained data that could be used for the analyses in this study. This resulted in a final subset containing 19 peer-reviewed articles and two preprints (*Yongchen et al., 2020*; *Du et al., 2020*; *To et al., 2020*; *Wölfel et al., 2020*; *Okba et al., 2020*; *Zhao et al., 2020*; *Haveri et al., 2020*; *Xiao et al., 2020*; *Jiang et al., 2020*; *Lee et al., 2020*; *Liu et al., 2020a*; *Long et al., 2020*; *Lou et al., 2020*; *Thevarajan et al., 2020*; *Xiang et al., 2020*; *Zhang et al., 2020a*; *Zhou et al., 2020*; *Young et al., 2020*; *Zhang et al., 2020b*; *Liu et al., 2020b*; *Zhang et al.,*

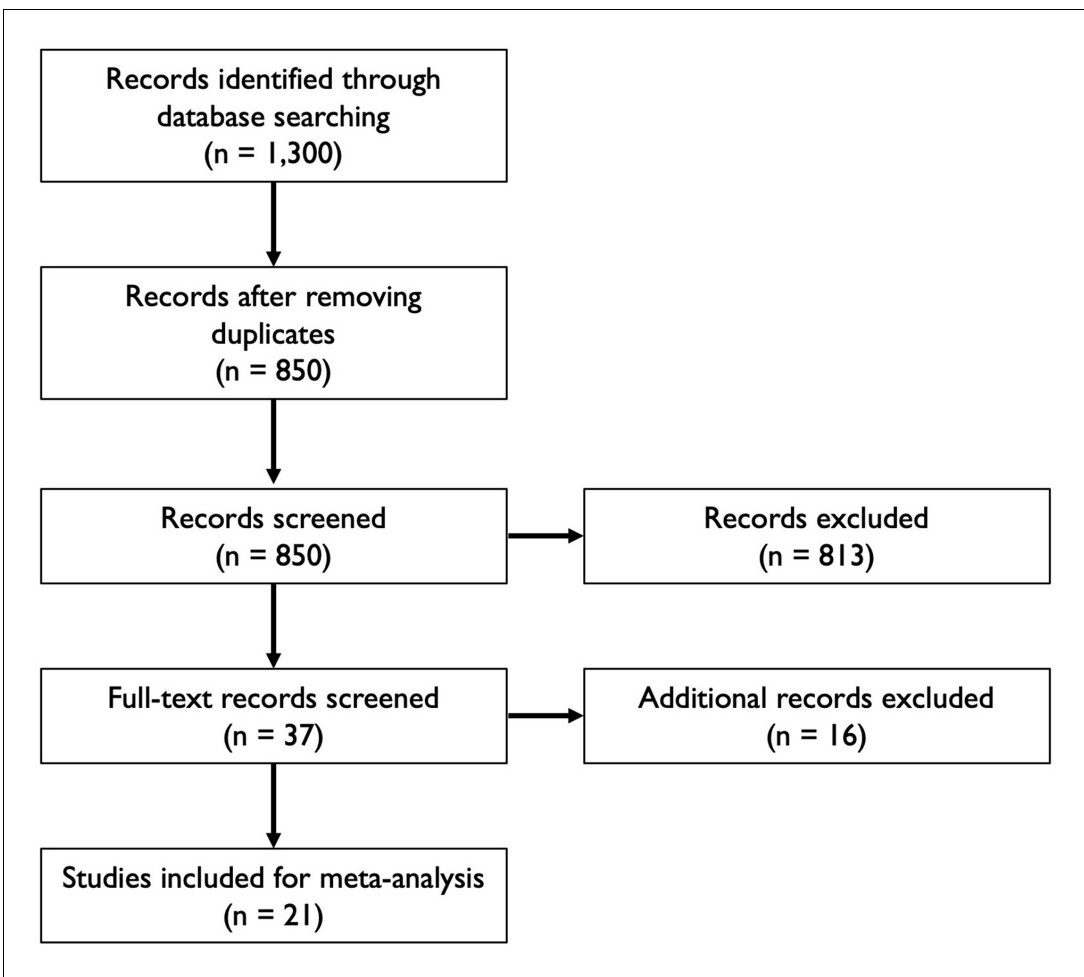

**Figure 5.** Flowchart illustrating the article selection process for the meta-analysis.

The online version of this article includes the following source data for figure 5:

**Source data 1.** Key features of articles used for analysis.

*2020c*; *Adams et al., 2020*; *Zou et al., 2020*). Note that initial article selection sample sizes are approximate due to the way in which Google Scholar reports the number of results. It was crucial for these searches to use Google Scholar in order to find preprints that are not included in databases such as Web of Science. *Figure 5—source data 1* provides an overview of all articles that were included for analysis, with key features noted. Analyses were done in parallel for a dataset excluding data from preprints, which did not change any qualitative results (not shown). The supplementary R code (*Source code 1*) includes the option to generate all results with or without data from preprints. Data were extracted from published material, and were digitized from figures when necessary using WebPlotDigitizer (*Rohatgi, 2019*). All data are available as *Source data 1*.

## Disease severity classification

Disease severity information was classified into three groups: asymptomatic/subclinical, mild/moderate, and severe/critical. Individuals were assigned a classification of asymptomatic/subclinical (N = 11) if they were referred to as 'healthy', 'having no symptoms related to COVID-19', or 'asymptomatic'. Inclusion criteria for classification as mild/moderate or severe/critical are based on definitions from the Centers for Disease Control and Prevention (*Centers for Disease Control and Prevention, 2019*), the Chinese National Health Commission (*Released by National Health Commission & National Administration of Traditional Chinese Medicine on March 3, 2020, 2020*), and the World Health Organization (*World Health Organization, 2020*). When disease severity was

not specified in the manuscript, patients who did not require supplemental oxygen therapy or transfer to the intensive care unit (ICU) were classified as mild/moderate, while those who did were classified as severe/critical.

## Estimating the distribution of seroconversion times

A major goal of this study is to estimate the means and variation of IgG and IgM seroconversion times (time between symptom onset and first antibody detection) for different assays, antigens, and disease severity. We developed a stepwise weighted bootstrapping procedure to do this using data on seroconversion times that have been reported in a diverse number of ways (from exact days to periods up to 22 days, and as raw results for one individual or means for groups of individuals). Our approach ensures that the best data (i.e. high-resolution data in the form of one specific seroconversion time for one individual) have the most influence on estimates of the means and standard deviations (sd) of seroconversion times.

The stepwise weighted bootstrapping procedure integrated all types of data that contain useful information about the timing of seroconversion of different antibodies in day(s) post-symptom onset. At each step, a distribution of observed possible seroconversion times was bootstrapped 50,000 times from repeated random sampling of individual seroconversion times from the dataset. At the end of each bootstrapping step $n$, a normal distribution was fitted to the obtained distribution of possible seroconversion times. This distribution was then used as prior information to weight sampling probability during the weighted bootstrapping procedure at step $n + 1$ (*Sms and Young, 2003*).

The first step used only the best available resolution of seroconversion data (i.e. reported for exact days, as opposed to a range of days) to bootstrap a distribution of observed possible seroconversion times. The following steps included all data for which the maximum reported seroconversion time range is the next one observed in the data (for up to maximum time range present in a dataset). For example, if a number of results was reported not as an exact time but as a period ranging 3 days (reported as such or as part of a time series), the data included in step 2 consist of results reported as exact days, and results reported as 3-day ranges. Bootstrapping in this case was again done through repeated random sampling of an individual. When that individual had a result reported as an exact time, that time was stored as a bootstrap sample. When that individual had a result reported as a time range, a time within that range was sampled, but importantly, the times within that range did not have the same probability of being sampled. This probability was determined by the normal distribution that was estimated after the preceding bootstrapping step. This ensured that the best available data have the largest contribution to the analysis, and data of lower resolution were used while taking into account the information contained in the higher resolution data. This stepwise procedure continued until data of all resolutions (i.e. including the largest reported seroconversion time periods) was bootstrapped.

Seroconversion times were sometimes reported as a mean time (± error) instead of an exact time or time period. In these cases, the standard deviation of time around the mean was calculated (using reported sample size and standard error), and a random time was drawn from this normal distribution. Some studies report seroconversion times for groups of individuals simultaneously. In this case, each individual group member was treated as a separate individual that can be sampled randomly. Data from cumulative seroconversion curves were incorporated by assigning the seroconversion time at which the curve increases to the number of individuals being reported to seroconvert at that time. In the bootstrapping procedure, each of these individuals could then be sampled in the same way as any other individual. Aside from increasing sample size (and hence the confidence in the estimates) and the density of the histogram/distribution, there were no significant differences between distributions estimated using different maximum time periods (*Figure 1—figure supplement 5*).

## Detection probability of IgG and IgM

The probability of detecting SARS-CoV-2 specific IgG or IgM in plasma or serum samples was estimated by integrating data on whether an individual tested positive or negative on a given day post-symptom onset. Data containing information on detection probability on a given day are reported in diverse ways, using different resolutions of sample size (from one individual to results reported for groups) or time (results reported on specific days or as a range of days). Additionally, time series

data from individuals sampled multiple times contain information about detection probability for times between measurements. These diverse data sources were integrated using different rules. When antibody levels were reported, the cut-off provided in the studies was used to determine the negative or positive status of samples. Individual results for a specific day were included as reported. When time was reported as a period, the midpoint time was used. When a proportion of positive samples was reported together with a sample size, the number of positive and negative samples were calculated and used as independent samples. When two samples that are part of a longitudinal time series showed the same result, the individual was assumed to have the same result for all times within the interval. When such samples had different results, the (interpolated) samples in the early half of the interval were assigned the same result as the first sample, and those in the later half were assigned the same result as the second sample. This procedure resulted in a dataset where each day post-symptom onset has a number of positive and negative observed samples that could be used to estimate a daily detection probability. Binomial exact confidence intervals of the means were calculated and shown.

## Detection probability of RNA

The probability of detecting RNA in upper and lower respiratory samples, and in fecal samples, was estimated using the same procedure used for IgG and IgM, but excluding the assumption that days in the interval between two samples of a time series have the same result, that is not including any interpolated samples. This was based on the fact that RNA detection has been observed to be highly variable (*Wölfel et al., 2020*; *Kucirka et al., 2020*). Respiratory sample types were classified as upper (saliva, naso- or oropharyngeal) or lower (sputum, tracheal aspirate, bronchoalveolar lavage) respiratory tract samples. As RT-PCR protocols based on different target sequences resulted in similar sensitivities (*Sethuraman et al., 2020*), all data were pooled for our analysis of detection probability.

## Antibody level kinetics

To characterize the kinetics of antibody levels, we fit models to all individuals for whom longitudinal data were available (i.e. at least three samples are available, one of which has to be positive). Our goal was to estimate the rate of increase, and the timing and magnitude of the peak antibody level. Assays, antigens and reporting units differed extensively between studies, so antibody levels were normalized by dividing the level of each sample in a study by the maximum value observed in that study. This allowed us to compare antibody level kinetic patterns between different studies. Antibody level normalization using scaling to a mean of zero and standard deviation of one resulted in the same patterns (results not shown). All time-series are shown in Appendix 1.

As there were no (or very limited) data available for the later phase of kinetics, when antibody levels decay from their peak, we focused on the early phase of antibody increase up to peak level. These early-phase dynamics follow a standard growth rate pattern, for which well-described functions are available. Of these functions, a three-parameter Gompertz function, $y(t) = ae^{-be^{-ct}}$, was an excellent candidate, as its three parameters correspond to clinically significant measures of antibody level ($y$) dynamics over time ($t$). The asymptote ($a$) corresponds to the peak level, displacement ($b$) corresponds to the seroconversion time, and growth rate ($c$) corresponds with the antibody level increase rate. Antibody levels ($y$ and $a$) were log-transformed.

We fit this function to the observed time series of normalized antibody levels using Bayesian Markov Chain Monte Carlo inference, using R-JAGS (*Plummer, 2019*). All parameters were fit separately for each individual, with the assumption that they arise from the same population-level distribution, which was implemented as a hierarchical Bayesian model with hyperpriors for each parameter.

Prior distributions:

> Peak titer mean ~Uniform(min = 0, max = 5)
> Peak titer standard deviation ~Gamma(shape = 1, rate = 1)
> Displacement mean ~Normal(mean = 100, sd = 10)
> Displacement standard deviation ~Uniform(min = 0, max = 200)
> Growth rate mean ~Uniform(min = 0, max = 5)
> Growth rate standard deviation ~Uniform(min = 0, max = 100)

Posterior means of the parameters were used for further analyses and for plotting. Data were combined into subsets depending on the measure of interest (assay, targeted antigen, disease severity). Six parallel chains with different starting values were run for 70,000 burn-in iterations, of which the first 20,000 were discarded (burn-in). Peak antibody level timing of an individual time series was approximated as the time at which the level reaches 95% of the maximum level ($a$). Results of parameters estimated using MCMC inference were reported as posterior means with 95% credible intervals (CrI). Statistical differences between estimated parameters were assessed by constructing the posterior distribution of the differences between the MCMC samples of the respective parameters (which were independent since they were estimated from different datasets), where the difference is considered significant when zero is not included in the 95% CrI.

All data preparation, cleaning, analysis and plotting was done in R version 3.6.1 (*R Development Core Team, 2019*) using packages ggplot2 (*Wickham, 2016*), dplyr (*Wickham et al., 2019*), readxl (*Wickham and Bryan, 2019a*), patchwork (*Pedersen, 2019*), binom (*Dorai-Raj, 2014*), tidyr (*Wickham and Henry, 2019b*) and ggridges (*Wilke, 2020*). Welch two-sample t-tests were used to test for differences between estimated distributions. All codes used to fit models and produce results have been provided in *Source code 1*.

## Acknowledgements

BB was supported by the European Commission Horizon 2020 Marie Sklodowska-Curie Actions (grant no. 707840). JOL-S and AG are supported by the Defense Advanced Research Projects Agency DARPA PREEMPT # D18AC00031 and the UCLA AIDS Institute and Charity Treks, and JOL-S and KCP are supported by the U.S. National Science Foundation (DEB-1557022), the Strategic Environmental Research and Development Program (SERDP, RC-2635) of the U.S. Department of Defense and the Cooperative Ecosystem Studies Unit Cooperative Agreement #W9132T1920006. The content of the article does not necessarily reflect the position or the policy of the U.S. government, and no official endorsement should be inferred.

## Additional information

### Funding

| Funder | Grant reference number | Author |
| --- | --- | --- |
| H2020 Marie Skłodowska-Curie Actions | 707840 | Benny Borremans |
| Defense Advanced Research Projects Agency | PREEMPT D18AC00031 | Amandine Gamble<br>James O Lloyd-Smith |
| UCLA AIDS Institute and Charity Treks | | Amandine Gamble<br>James O Lloyd-Smith |
| National Science Foundation | DEB-1557022 | KC Prager<br>James O Lloyd-Smith |
| U.S. Department of Defense | Strategic Environmental Research and Development Program RC-2635 | KC Prager<br>James O Lloyd-Smith |
| Cooperative Ecosystem Studies Unit | Cooperative Agreement W9132T1920006 | KC Prager<br>James O Lloyd-Smith |

The funders had no role in study design, data collection and interpretation, or the decision to submit the work for publication.

### Author contributions

Benny Borremans, Conceptualization, Data curation, Software, Formal analysis, Supervision, Investigation, Visualization, Methodology, Writing - original draft, Writing - review and editing; Amandine Gamble, Conceptualization, Data curation, Formal analysis, Methodology, Writing - review and editing; KC Prager, Resources, Data curation, Investigation, Methodology, Writing - review and editing; Sarah K Helman, Resources, Data curation, Investigation, Writing - review and editing; Abby M

McClain, Resources, Data curation, Formal analysis, Investigation, Writing - review and editing; Caitlin Cox, Resources, Investigation, Writing - review and editing; Van Savage, Investigation, Methodology, Writing - review and editing; James O Lloyd-Smith, Conceptualization, Investigation, Methodology, Writing - review and editing

### Author ORCIDs
Benny Borremans  https://orcid.org/0000-0002-7779-4107
KC Prager  https://orcid.org/0000-0003-0669-0754
Abby M McClain  http://orcid.org/0000-0001-5000-4198
James O Lloyd-Smith  http://orcid.org/0000-0001-7941-502X

### Decision letter and Author response
Decision letter https://doi.org/10.7554/eLife.60122.sa1
Author response https://doi.org/10.7554/eLife.60122.sa2

## Additional files

### Supplementary files
- Source code 1. R code.
- Source data 1. Extracted data used for meta-analysis.
- Transparent reporting form

### Data availability
All data are available in Source data 1.

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

# Appendix 1

## Antibody level kinetics MCMC outputs

The figures below show the antibody data used for the different antibody/assay/antigen datasets. The IDs shown in the figure legend correspond with the individual identification provided in the accompanying spreadsheet. Antibody levels by days post symptom onset are shown for each individual (top panel). Posterior growth rate, displacement and peak antibody level are shown (middle panels), with posterior mean (bold red line), and 95% credible intervals (dashed lines). Observed antibody levels (bottom-left panel) and fitted functions (bottom-middle panel) are shown for each individual, in addition to the overall mean (black dashed line). Finally, the posterior mean antibody level and 100 randomly selected posterior fits within the 95% credible interval are shown (bottom-right panel). Chain convergence was assessed using the Gelman-Rubin diagnostic, and was one for all chains.

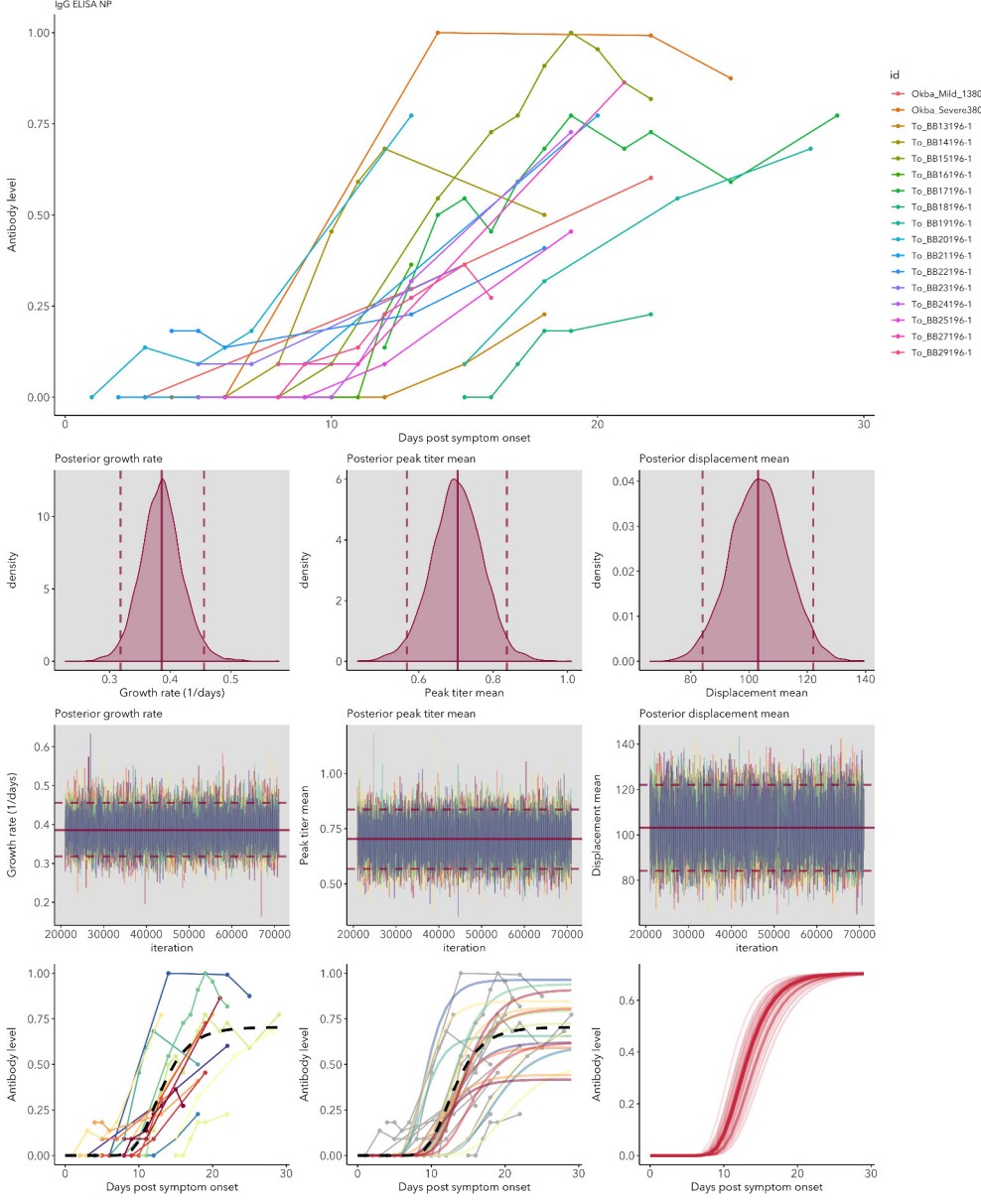

**Appendix 1—figure 1.** IgG ELISA-NP fitted antibody kinetics.

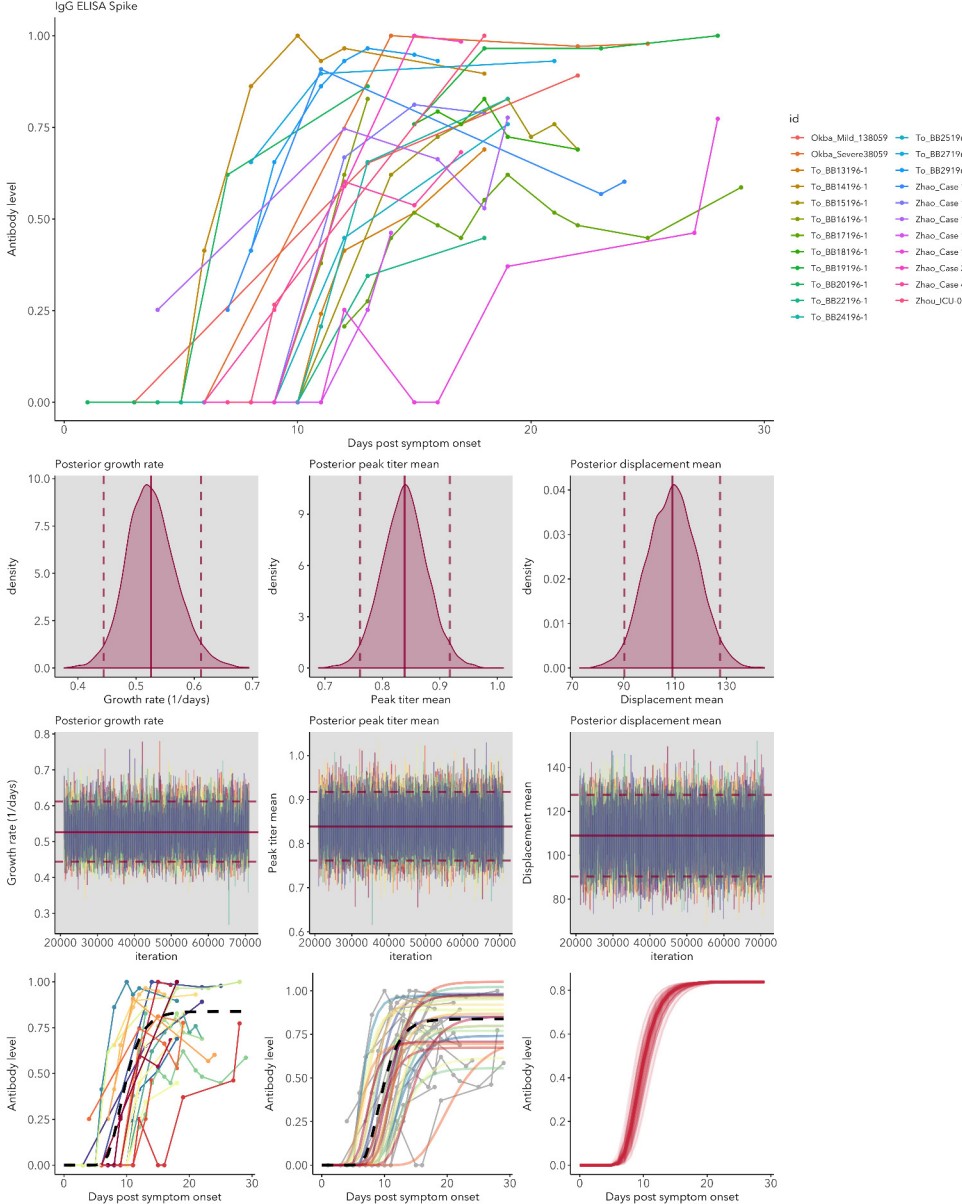

**Appendix 1—figure 2.** IgG ELISA-Spike fitted antibody kinetics.

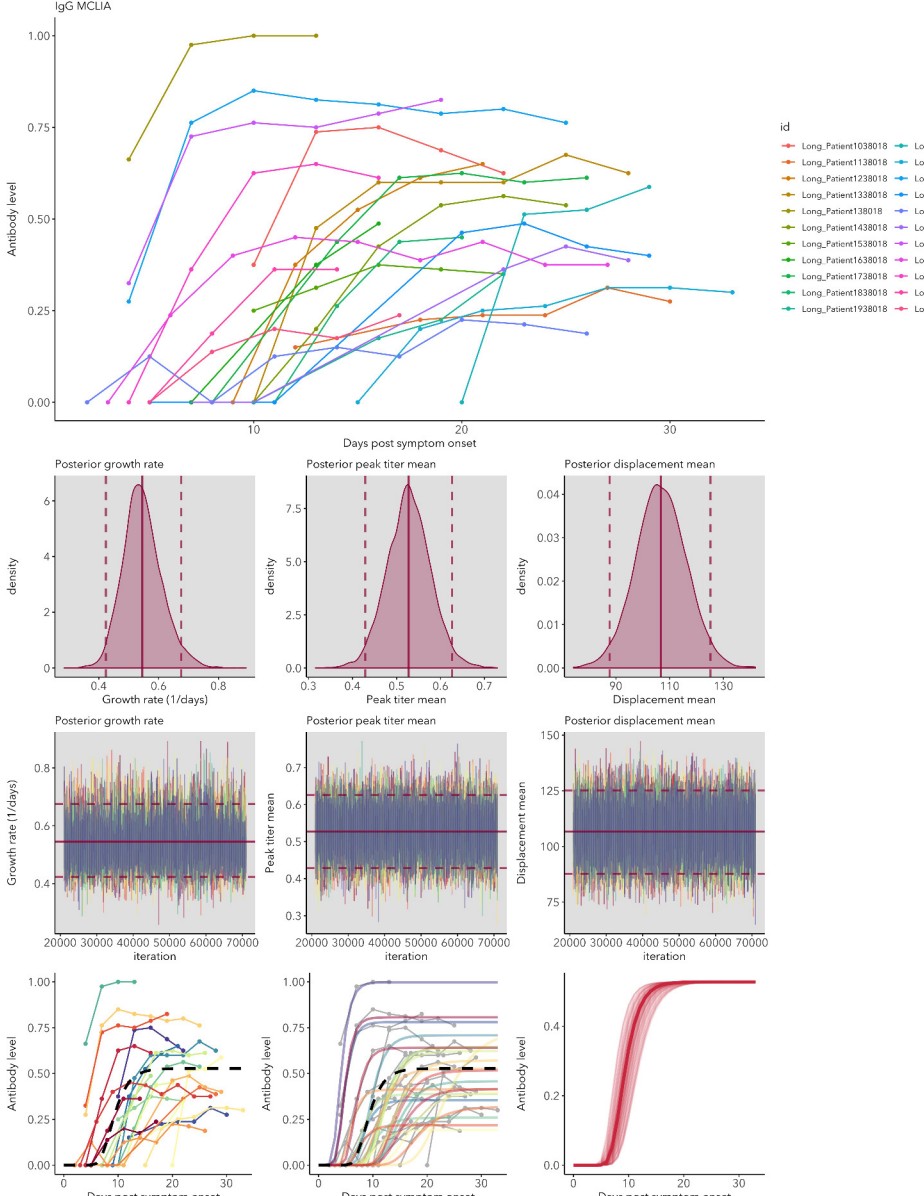

**Appendix 1—figure 3.** IgG MCLIA fitted antibody kinetics.

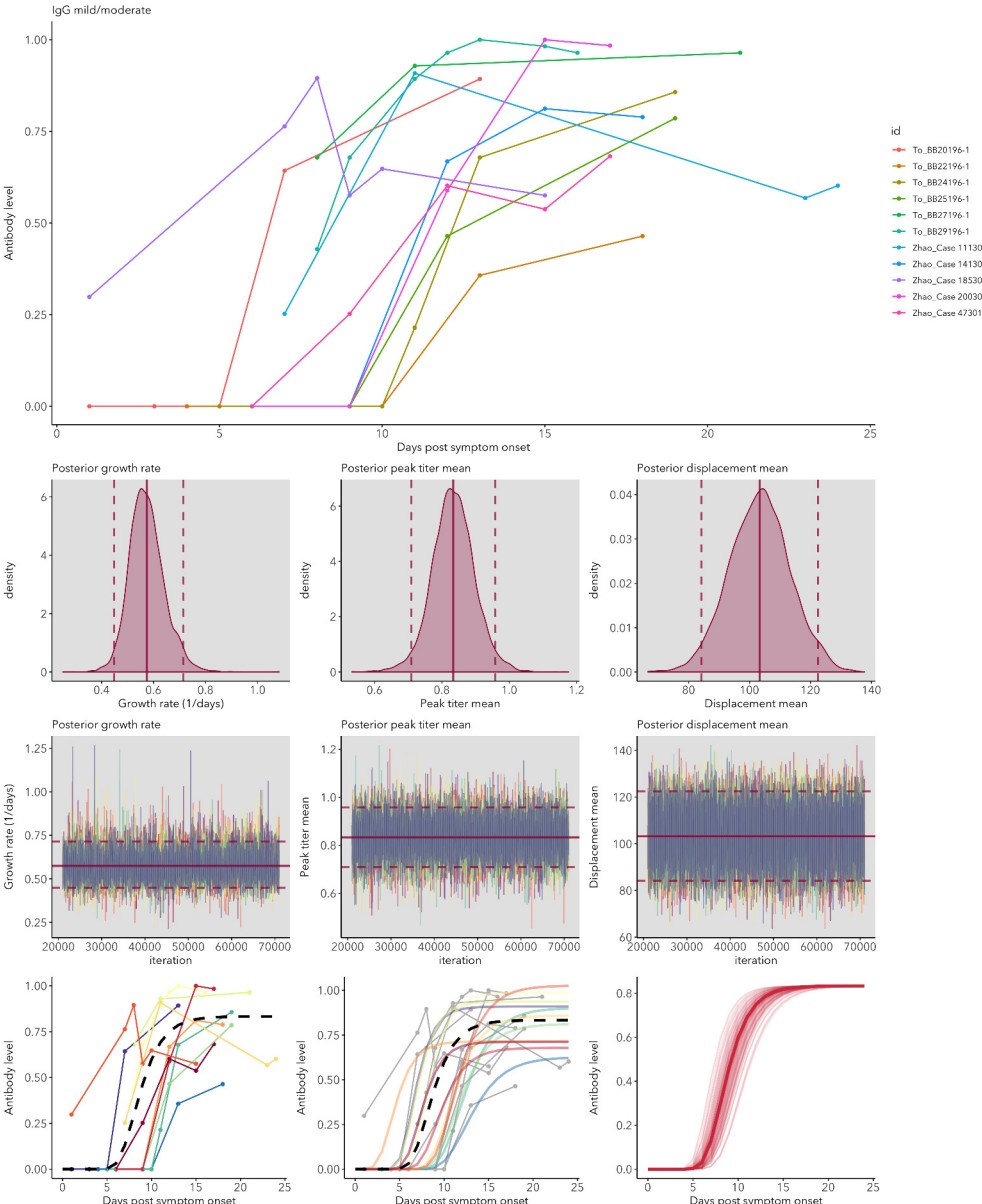

**Appendix 1—figure 4.** IgG mild/moderate cases fitted antibody kinetics.

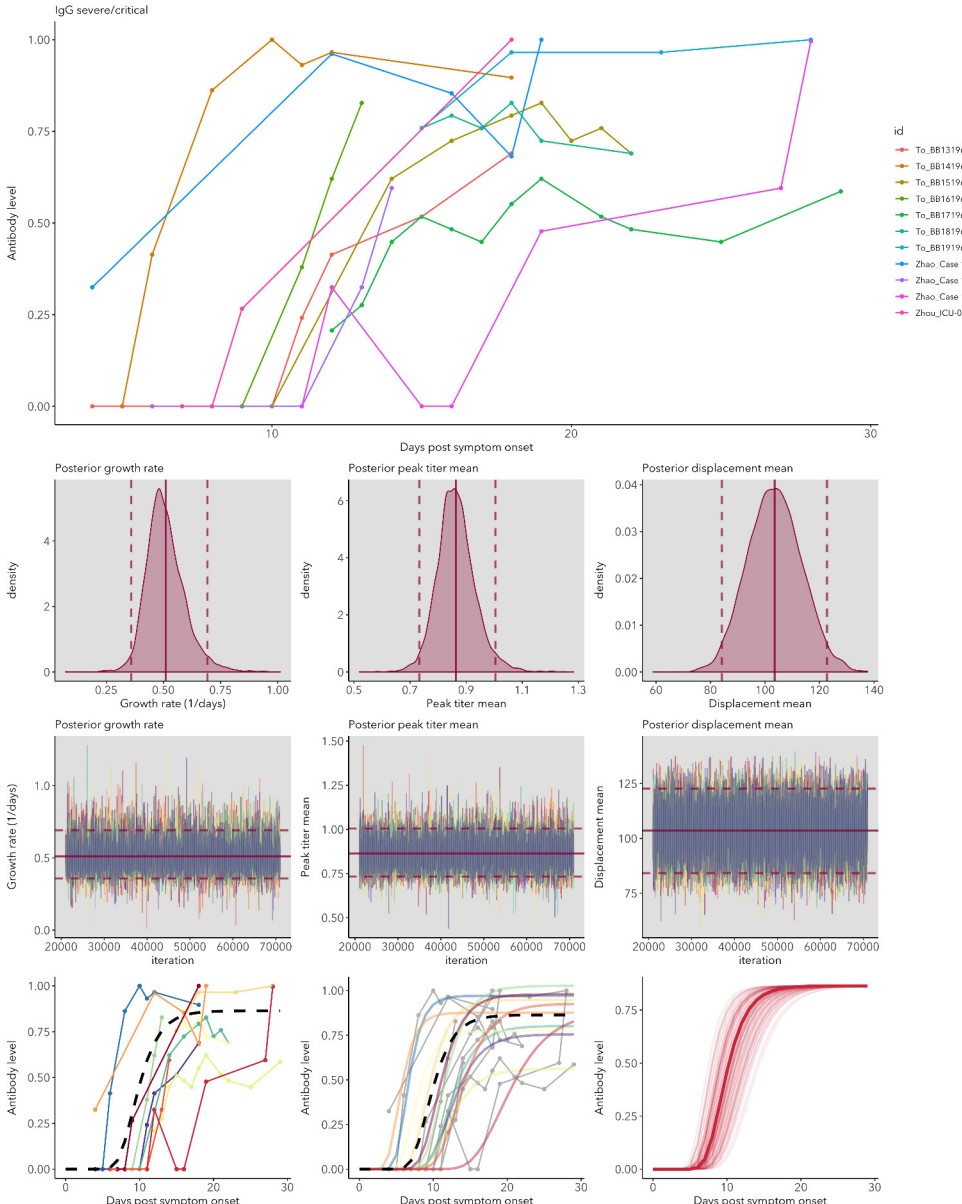

**Appendix 1—figure 5.** IgG severe/critical cases fitted antibody kinetics.

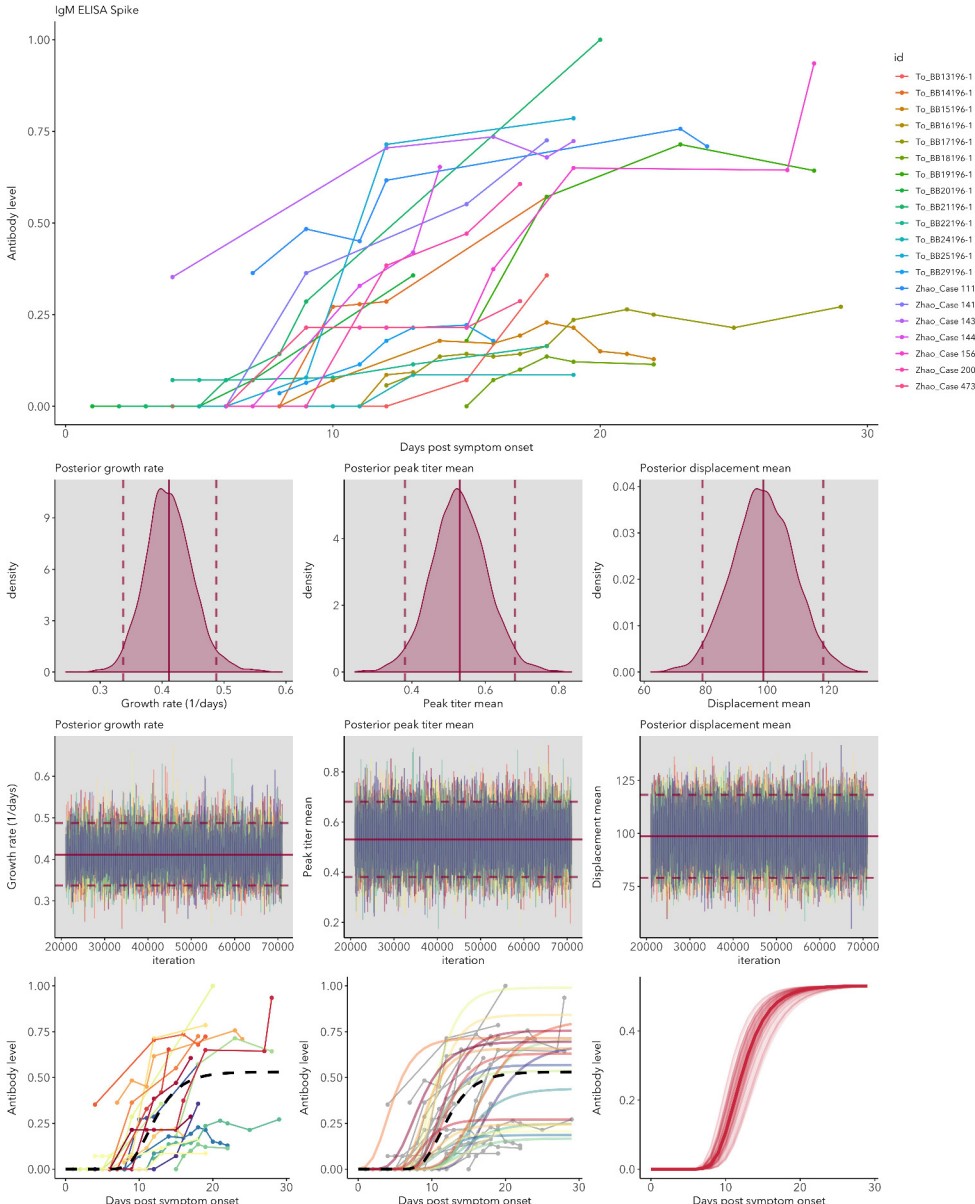

**Appendix 1—figure 6.** IgM ELISA-Spike fitted antibody kinetics.

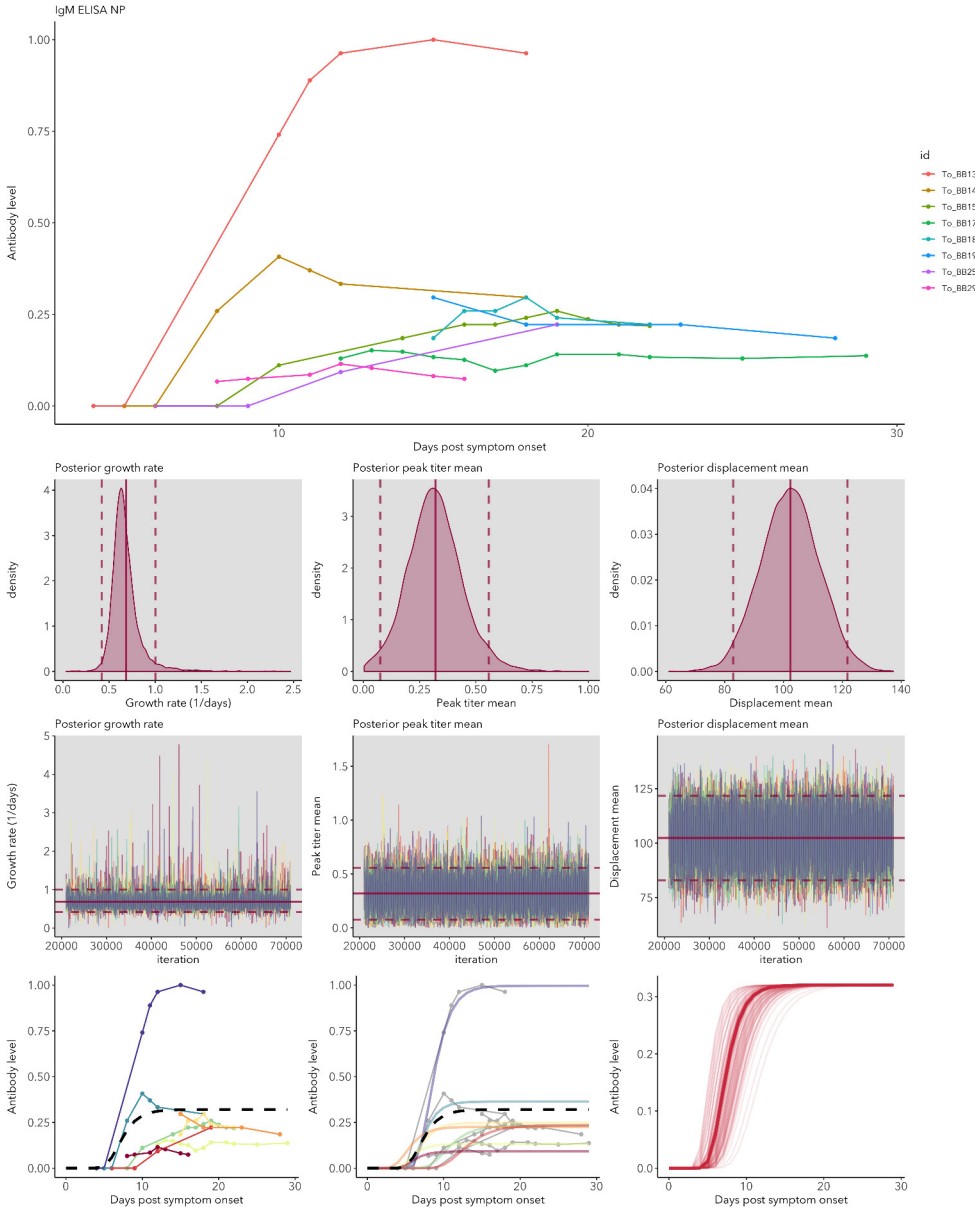

**Appendix 1—figure 7.** IgM ELISA-NP fitted antibody kinetics.

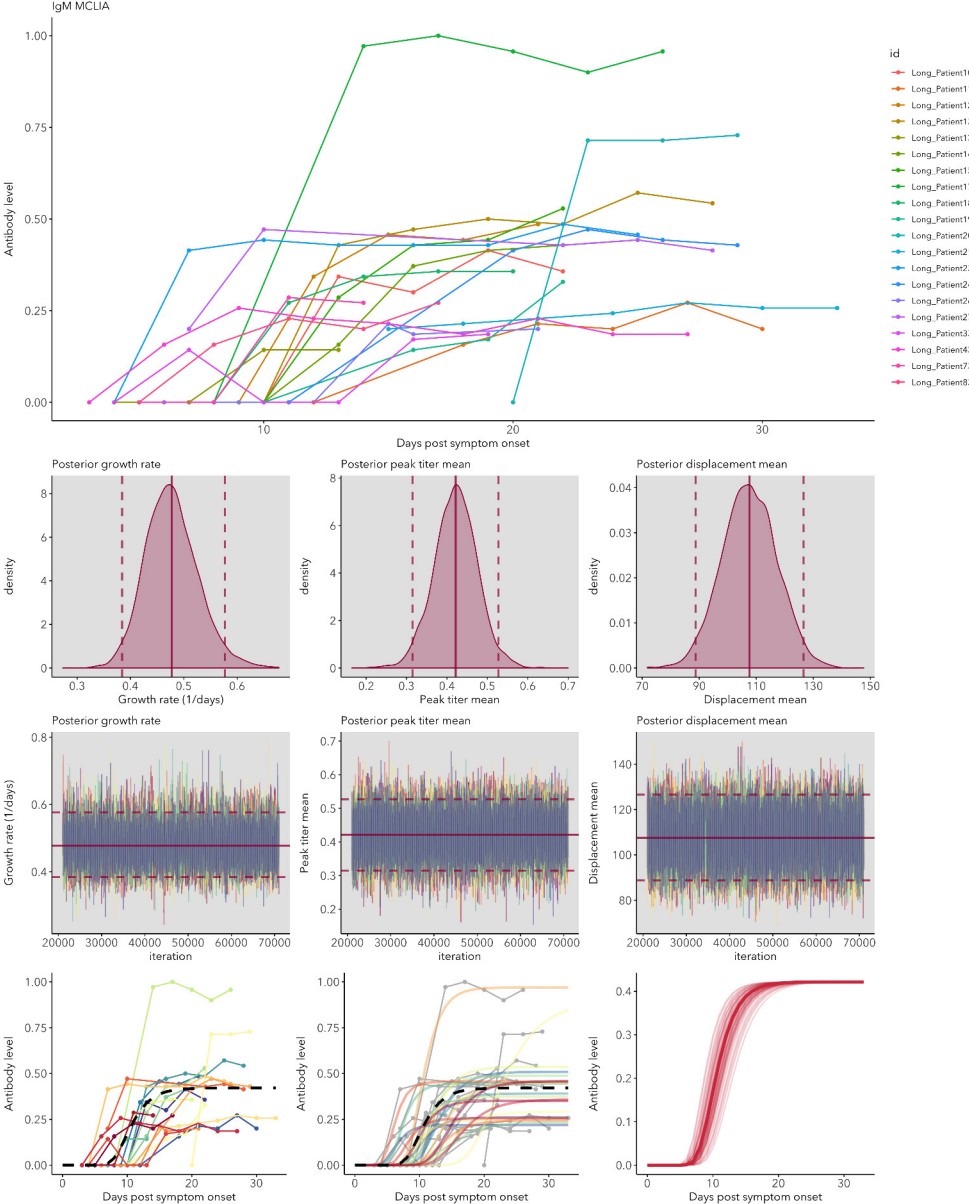

**Appendix 1—figure 8.** IgM MCLIA fitted antibody kinetics.

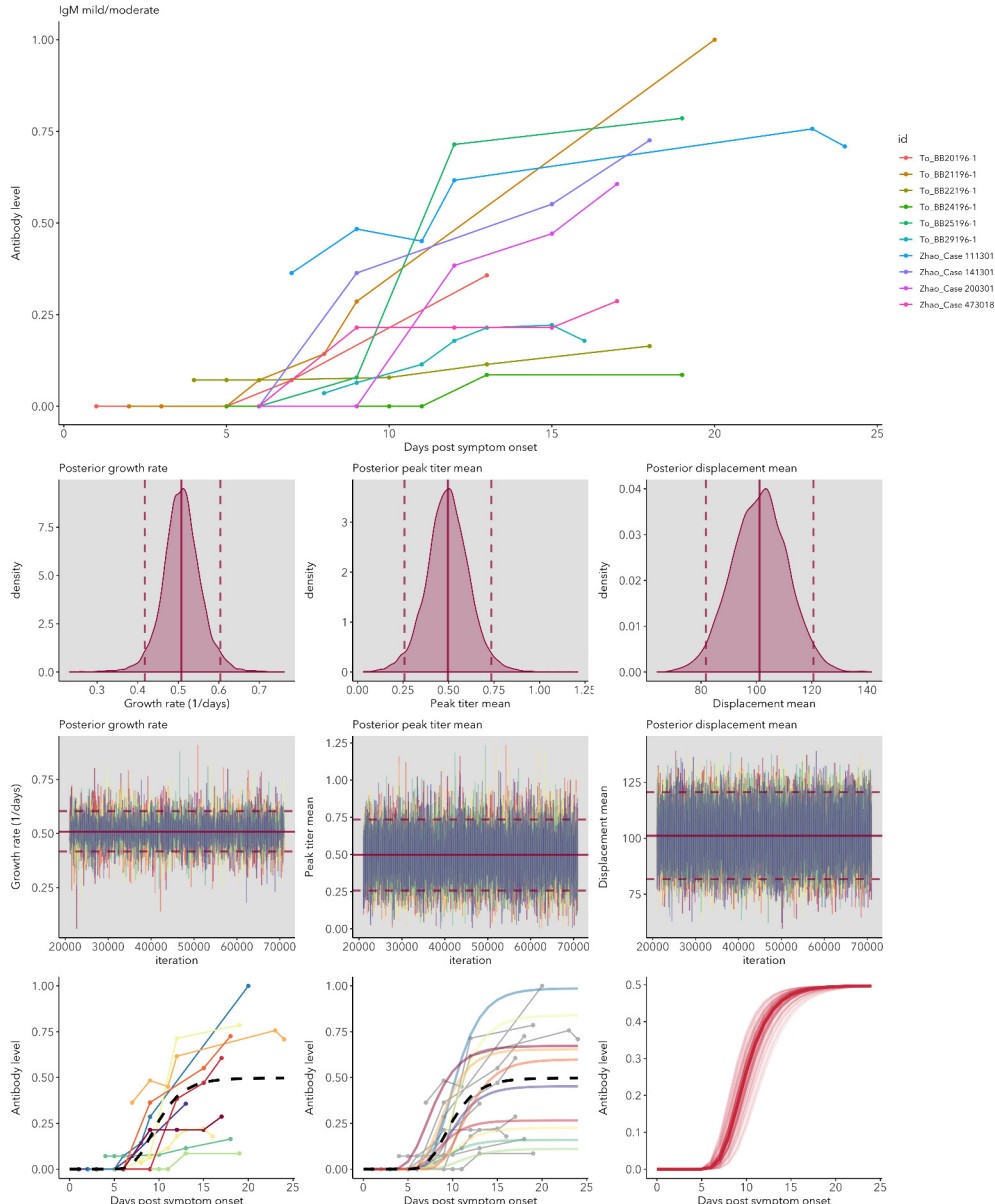

**Appendix 1—figure 9.** IgM mild/moderate cases fitted antibody kinetics.

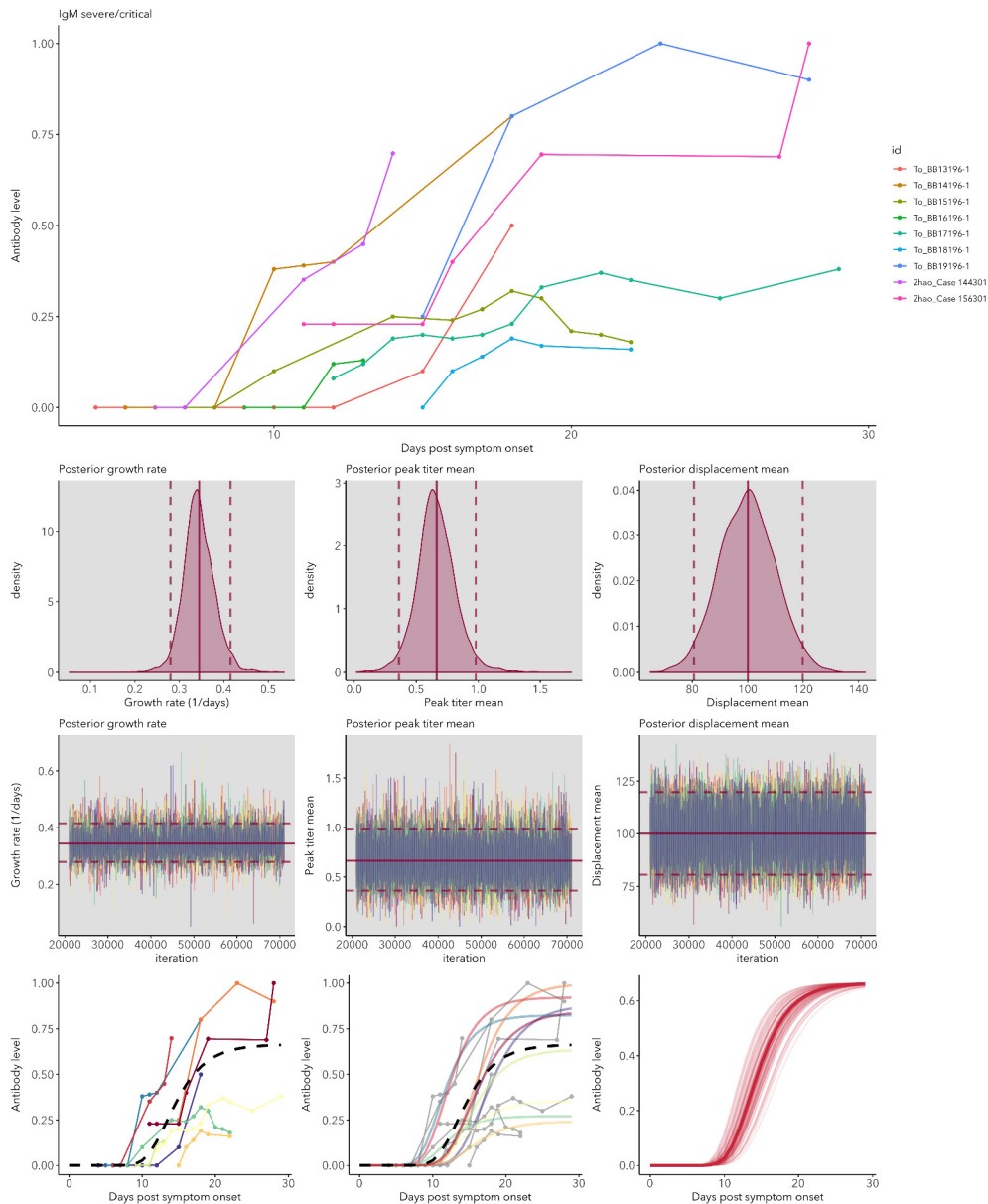

**Appendix 1—figure 10.** IgM severe/critical cases fitted antibody kinetics.

