## [Decision Letter]

**Acceptance summary:**

This manuscript presents an important and timely assessment of current data on antibody kinetics and RNA shedding during SARS-CoV-2. The researchers used innovative estimating procedures that draw from multiple data streams and data sources. The characterization of detection probabilities of different tests over time is topical and very relevant for clinical use, public health and modeling and other research.

**Decision letter after peer review:**

Thank you for submitting your article "Quantifying antibody kinetics and RNA shedding during early-phase SARS-CoV-2 infection" for consideration by *eLife*. Your article has been reviewed by three peer reviewers, one of whom is a member of our Board of Reviewing Editors, and the evaluation has been overseen by Miles Davenport as the Senior Editor. The following individual involved in review of your submission has agreed to reveal their identity: Michael J Mina (Reviewer #3).

The reviewers have discussed the reviews with one another and the Reviewing Editor has drafted this decision to help you prepare a revised submission.

Summary:

This manuscript presents an important and timely assessment of current data on antibody kinetics and RNA shedding during SARS-CoV-2. All reviewers expressed that they greatly enjoyed reading the paper and thought it is exactly the type of data integration that needs to be taking place in this fast paced and sparse data world that is COVID-19. However, there were some methodological concerns that should be addressed in a revision as we felt they may influence the validity of the estimates in this paper:

Essential revisions:

1) Reporting of the review component: This paper is missing some reporting elements that are usual in reviews of the literature and required for full transparency. Please provide: a) the full electronic search strategy for at least one database in appendix, with search terms used; b) a flowchart of the study selection process, with inclusions and exclusions; c) a summary table of all included studies, providing an overview of their main characteristics (ex. assay, antigen, countries, age range of participants, sample sizes, and maximum follow-up times) ; d) the full list of citations of all included studies. While this paper is not a formal systematic review, some guidance on how to present the elements listed above can be found in the PRISMA guidelines.

2) Chain convergence: Looking at the plots in the supplementary information, the MCMC chains have not converged. This means that the results on the fitting of antibody levels cannot be reliably interpreted and casts doubt on the validity of estimates from these analyses. There are several individuals for which we observe very poor fits. Perhaps visualizing the uncertainty in the model estimates of antibody levels would clarify. The chains need to be run again. Calculation of the Gelman-Rubin convergence statistic would provide evidence of chain convergence. The reviewers recommend running the chains for longer and re-tuning the proposal distributions in order to achieve chain convergence.

---

## [Author Response]

Essential revisions:1) Reporting of the review component: This paper is missing some reporting elements that are usual in reviews of the literature and required for full transparency. Please provide: a) the full electronic search strategy for at least one database in appendix, with search terms used; b) a flowchart of the study selection process, with inclusions and exclusions; c) a summary table of all included studies, providing an overview of their main characteristics (ex. assay, antigen, countries, age range of participants, sample sizes, and maximum follow-up times) ; d) the full list of citations of all included studies. While this paper is not a formal systematic review, some guidance on how to present the elements listed above can be found in the PRISMA guidelines.

We agree that more information about the selection process would be valuable, and have added all information as suggested (subsection “Estimating the distribution of seroconversion times”). We have also added a flowchart (Figure 5) and a table listing the key features of the selected articles (Figure 5—source data 1).

2) Chain convergence: Looking at the plots in the supplementary information, the MCMC chains have not converged. This means that the results on the fitting of antibody levels cannot be reliably interpreted and casts doubt on the validity of estimates from these analyses. There are several individuals for which we observe very poor fits. Perhaps visualizing the uncertainty in the model estimates of antibody levels would clarify. The chains need to be run again. Calculation of the Gelman-Rubin convergence statistic would provide evidence of chain convergence. The reviewers recommend running the chains for longer and re-tuning the proposal distributions in order to achieve chain convergence.

We fully agree, and have re-done all MCMC fitting using a better method (JAGS instead of R Metropolis-Hastings MCMC).

Chains for all parameters now converge nicely, with clean posterior distributions.

Additionally, the much improved model fits have motivated us to improve parameter estimation by implementing individual growth rates (as opposed to one population-level growth rate previously).

These improvements did not change any qualitative results, and now allowed all statistical comparisons of model parameters to be done fully based on posterior distributions that take into account parameter uncertainty.

These changes resulted in the following edits in the main text and supplementary information:

– Updated description of MCMC model fitting in the main text.

– Updated antibody kinetics results in the main text

– Updated MCMC results in Figure 3—figure supplements 1-10.

Unfortunately a bug was found in the code in the process of improving the models, which previously resulted in wrong subsetting of IgM results of mild cases.

As a result, we now find a marginally significant difference in IgM antibody level growth rates and peak level timing between mild and severe cases.

This did not have a major impact on the overall conclusions however, as the difference was not large, and seroconversion as well as antibody detection patterns that are based on much larger datasets do not provide evidence for a general effect of disease severity on antibody patterns.

We edited the relevant section in the Discussion accordingly: "Here, we did not detect any significant effects of disease severity on antibody patterns, with the single exception that we estimated a lower rate of IgM increase in severe/critical cases relative to mild/moderate cases", and "Our findings do not support the idea that severe cases seroconvert faster. Indeed, the only significant effect of severity in our analyses is that the inferred growth rate of IgM levels is slower for severe/critical cases. It is not clear whether this reflects a relevant biological difference, considering that all other parameters do not differ among disease severity categories. The consensus patterns from our meta-analysis suggest that any interaction between disease severity and antibody response must be subtle and sensitive to other sources of variation, explaining the inconsistencies seen across studies".